# Advances in mRNA-Based Cancer Vaccines

**DOI:** 10.3390/vaccines11101599

**Published:** 2023-10-16

**Authors:** Ling Ni

**Affiliations:** Institute for Immunology and School of Medicine, Tsinghua University, Medical Research Building, No. 30 Haidian Shuangqing Road, Beijing 100084, China; lingni@tsinghua.edu.cn

**Keywords:** cancer, immunotherapy, cancer vaccine, mRNA cancer vaccine, neoantigen discovery, adjuvant identification, delivery materials

## Abstract

Cancer is a leading cause of death worldwide, accounting for millions of deaths every year. Immunotherapy is a groundbreaking approach for treating cancer through harnessing the power of the immune system to target and eliminate cancer cells. Cancer vaccines, one immunotherapy approach, have shown promise in preclinical settings, but researchers have struggled to reproduce these results in clinical settings. However, with the maturity of mRNA technology and its success in tackling the recent coronavirus disease 2019 (COVID-19) pandemic, cancer vaccines are expected to regain attention. In this review, we focused on the recent progress made in mRNA-based cancer vaccines over the past five years. The mechanism of action of mRNA vaccines, advancements in neoantigen discovery, adjuvant identification, and delivery materials are summarized and reviewed. In addition, we also provide a detailed overview of current clinical trials involving mRNA cancer vaccines. Lastly, we offer an insight into future considerations for the application of mRNA vaccines in cancer immunotherapy. This review will help researchers to understand the advances in mRNA-based cancer vaccines and explore new dimensions for potential immunotherapy approaches.

## 1. Introduction

Cancer indeed is a serious and complex disease that can result in significant morbidity and mortality [1]. Many cancers are caused by somatic mutations, which are changes in the DNA of cells that accumulate over time due to various factors such as exposure to carcinogens, genetic predisposition, and other environmental influences [2,3,4]. One of the remarkable aspects of cancer is that these mutations can lead to the production of abnormal proteins, some of which are unique to cancer cells and not found in normal cells. These abnormal proteins, also known as neoantigens, are recognized as foreign or “non-self” by the immune system [5,6,7,8,9]. Most neoantigens are unique to an individual patient, but there are some neoantigens that are shared among many patients. These shared neoantigens are generally driver mutations that disrupt the functions of key proteins such as p53 and KRAS. Targeting these shared neoantigens might be faster and more affordable by mass production due to their presence in a high portion of cancers. In a healthy state, the immune system’s surveillance mechanisms recognize and eliminate such foreign entities, including pathogens and abnormal cells such as cancer cells.

However, cancer cells often develop mechanisms to evade immune recognition and destruction [10]. This is where immunotherapy comes into play. Immunotherapy is a groundbreaking approach for treating cancer by harnessing the power of the immune system to target and eliminate cancer cells [11]. There are different types of immunotherapies, and they all aim to enhance the immune response against cancer cells. Immunotherapy has indeed revolutionized cancer treatment. It offers new hope for patients who may not respond well to traditional treatments, such as chemotherapy or radiation therapy. It has shown impressive results in various types of cancers, leading to prolonged survival and sometimes even complete remission in patients who previously had limited treatment options [10].

One notable approach of immunotherapy is immune checkpoint blockade. Immune checkpoint molecules are regulatory proteins that help maintain a balance in the immune system and prevent excessive immune activation [12]. However, cancer cells can exploit these checkpoints to evade immune attacks. Drugs known as immune checkpoint inhibitors (ICIs) can block these inhibitory signals, unleashing the immune system’s ability to recognize and attack cancer cells [12]. Adoptive cell therapy (ACT) is another immunotherapy approach. For example, chimeric antigen receptor T cells (CAR-T cells) are engineered to express receptors that specifically target cancer cells, and they have shown remarkable success in treating certain types of blood cancers [13].

Cancer vaccines, another immunotherapy approach, have shown promise in preclinical settings, but researchers have struggled to reproduce these results in clinical settings. The immune response induced by cancer vaccines is insufficient to overcome the immunosuppressive tumor microenvironment (TME). In addition, cancer cells evade immune surveillance through multiple mechanisms, including restricting neoantigen recognition, suppressing the immune system, and inducing T-cell exhaustion, which represent a major roadblock for the development and treatment with cancer vaccines. Provenge, licensed by the Food and Drug Administration (FDA) in 2010, is the only licensed tumor vaccine for treating asymptomatic or mild symptomatic castration-resistant prostate cancer [14]. However, Provenge faded out of the market due to its high price and low efficacy with median overall survival of 25.9 and 21.4 months in the Provenge and placebo groups, respectively [15]. Since then, the development of cancer vaccines has stagnated, and several trials have failed one after the other. Fortunately, with the maturity of mRNA technology, tumor vaccines are expected to stop failing and regain attention.

The landscape of mRNA technology research is expanding rapidly, with over 600 systematic reviews available as of this writing. These reviews encompass a wide range of topics and applications, shedding light on the status and future directions of this transformative field. For example, Zhao et al. performed a pioneering global bibliometric analysis of publications related to therapeutic cancer vaccines ranging from 2013 to 2022 via VOSviewer and CiteSpace to study the existing state of research and potential trends in this field [16]. One comprehensive review delved into the immunotherapy strategies and underlying mechanisms related to mRNA vaccines in the context of gastrointestinal tumors [17]. Tojjari et al. undertook an extensive examination of various cancer vaccine types, including dendritic cell-based, viral vector-based, peptide-based, and mRNA and DNA vaccines, as well as their potential applications in hepatocellular carcinoma management [18]. Al Fayez et al. summarized various types of mRNA-based vaccines, illuminating their underlying mechanisms and discussing strategies to optimize the development while addressing existing limitations [19]. Han and colleagues conducted a review of the diverse applications of synthetic coding and non-coding RNAs, such as mRNA, miRNA antisense oligonucleotides, and circular RNA, in the realm of therapeutic development [20]. Liu et al. provided a comprehensive overview of in vitro transcribed mRNA-based therapeutics tailored for cancer treatment [21]. Wang et al. curated insights from clinical trials involving mRNA cancer vaccines [22]. These systematic reviews collectively contribute to our understanding of mRNA technology’s multifaceted potential and its role in shaping the future of cancer research and therapy.

In this review, we focused on the recent progress made in mRNA-based cancer vaccines over the past five years. We discussed the mechanism of action of mRNA vaccines. We also reviewed advancements in neoantigen discovery, adjuvants’ identification, and delivery methods. Additionally, we provided a detailed overview of recent clinical trials involving mRNA tumor vaccines and contemplated future considerations for the application of mRNA vaccines in cancer immunotherapy.

### mRNA Cancer Vaccines and Their Mechanism of Action

In the recent pandemic, two mRNA coronavirus disease 2019 (COVID-19) vaccines (mRNA-1273 and BNT162b2) were approved by the FDA [23,24], and they helped humans tackle the pandemic through immunization. Both mRNA-1273 and BNT162b2 consist of mRNA-encoding spike proteins stabilized in prefusion conformation. The spike proteins allow SARS-CoV-2 to enter host cells by binding to its putative receptors, and it has been the main vaccine target to prevent viral infection. The success of these two mRNA vaccines offers considerable hope for future mRNA vaccines, accelerates mRNA vaccine technology, and promotes the biopharmaceutical industry.

mRNA vaccines have emerged as a promising platform for cancer vaccines. Compared with conventional vaccine platforms, the mRNA vaccine advantages include cost-effective manufacturing, safe administration, high potency, and rapid development potential [22,24]. However, mRNA vaccines are unstable and inefficient when using in vivo delivery, which might limit their applications. Several investigations into appropriate mRNA structure modifications, such as codon optimization, nucleotide modification, self-amplifying mRNA, etc., have been conducted to overcome these issues [25]. Formulation methods (i.e., lipid nanoparticles (LNPs), polymers, peptides, etc.) have also been studied to boost the efficacy of mRNA vaccines [25].

mRNA vaccines work by introducing a single-stranded molecule that encodes viral proteins or neoantigens. Once injected, the mRNA cancer vaccines are delivered into the cytoplasm of antigen-presenting cells, such as dendritic cells (DCs), where they are translated into neoantigen proteins. In this step, they stimulate DCs via Toll-like receptors (TLRs) 7 and 8 [26,27]. Subsequently, mature DCs are transported to lymph nodes to generate remarkable humoral and cellular immunity [28,29]. Compared with conventional vaccines, mRNA vaccines induce a robust type I interferon response and are effective at eliciting CD8^+^ T-cell responses in humans [27], and this plays a key role in the eradication of tumors (Figure 1).

## 2. Neoantigen Discovery for mRNA Cancer Vaccines

Neoantigen identification has gained escalating significance in the realm of immunotherapies. However, identifying suitable neoantigens that can be targeted for the development of mRNA vaccines remains a significant challenge. Tretter et al. presented an extensive multi-omics dataset encompassing 32 patients spanning 25 diverse tumor types [30], with the aim of facilitating a proteogenomic-based neoantigen discovery. Their investigation unveiled intriguing findings: shared DNA and RNA variants, along with tumor-associated peptides, were identified across patients, demonstrating an independence from the specific tumor type. Remarkably, a majority of the identified neoantigens predominantly originated from RNA sources rather than DNA sources in most of the patients. These discoveries have emphasized the feasibility of employing proteogenomic approaches for neoantigen identification within a cross-entity cohort. Moreover, the prevalence of neoantigens derived from RNA sources suggests their potential as highly relevant targets in the pursuit of immunotherapy development [30]. These findings, however, require confirmation in a large patient cohort.

Cancer/testis antigens (CTAs) are a type of tumor antigen expressed in testis, placental tissues, and numerous cancer tissues. CTAs are currently the dominating non-mutated targets in the development of mRNA cancer vaccines. The human genome encodes more than 200 CTAs that can be measured by serology and expression levels. Lung cancer, bladder cancer, and skin cancer highly express CTAs [31], and they could be promising candidates for designing mRNA vaccines in the treatment of these three cancers. A proof-of-concept study should be conducted to evaluate the efficacy of mRNA-encoding CTAs in the targeted patient population.

In the context of colorectal cancer (CRC), increased expression levels of troponin T1, thrombospondin, biglycan, follistatin like 3, NADPH oxidase 4, and collagen triple helix repeat containing 1 were related with unfavorable overall survival outcome by a survival analysis. Additionally, these six tumor antigens were related with the antigen-presenting cell (APC) infiltration in CRC patients [32]. These findings indicated that these six tumor antigens could be potential neoantigens for designing an effective mRNA vaccine in the treatment of suitable CRC patients. 

Four primary molecular subtypes of breast cancer (BC) are defined in large part by hormone receptors (HR) and human epidermal growth factor receptor 2 (HER2), namely, luminal A (HR^+^/HER2^−^), luminal B (HR^+^/HER2^+^), HER2^+^, and triple-negative BC (HR^−^/HER2^−^) [33]. Triple-negative BC patients receive poor prognoses, and it is resistant to chemotherapy, insensitive to radiotherapy, and has a high prevalence of adverse drug reactions. Li et al. analyzed 1104 BC samples, of which 38.7% were luminal A, 17.1% luminal B, 6.1% HER2^+^, 12.8 triple-negative, and the remaining sample subtypes were not applicable. Potential BC-associated neoantigens for mRNA vaccines and populations suitable for mRNA vaccination were studied [34]. They identified three tumor-associated antigens (TAAs), such as interferon regulatory factor 1 (IRF1), the CD74 molecule, and proteasome activator subunit 2 [34]. Furthermore, three immune subtypes, namely, Clusters A, B, and C, were identified and Cluster B patients had a TME that was beneficial to immunotherapy. These subtypes also displayed distinct expression profiles of their immune cell death-promoting genes, immune checkpoints, and responses to ICI treatment [34]. These discoveries shed light on promising avenues for the mRNA BC vaccine development. The best efficacy for mRNA BC cancer vaccines can be achieved with multiple TAAs in suitable patients. These three TAAs [34] should be further confirmed in a large triple-negative BC patient cohort.

Wang et al. discovered adaptor-related protein complex 2 subunit sigma 1 (AP2S1), prolyl-3 hydroxylase family member 4 (P3H4), and rac family small GPTase 3 (RAC3) as potential tumor-specific antigens (TSAs) for bladder cancer on the basis of their expression levels and associations with prognoses when analyzing 165 primary bladder cancer tissue samples across four stages I-IV (103 superficial and 62 invasive) [35]. Moreover, based on immune-related gene expression profiles, three immune subtypes, namely, BCS1, 2, and 3, were classified. Patients categorized under the BCS2 subtype were identified as having an “immune cold” profile, marked by an increase in immunogenic cell death modulators. In contrast, patients belonging to the BCS1 and BCS3 subtypes were deemed “immune hot” due to the upregulation of immune checkpoints. Intriguingly, those falling into the BCS2 subtype displayed more favorable prognoses compared to patients with the other subtypes, making them potential candidates for mRNA vaccines, and it was determined that patients with the BCS1 and BCS3 subtypes may benefit from ICI therapy [35]. However, Zhang et al. used different methods to identify insulin-like growth factor 2 MRNA binding protein 2 (IGF2BP2) and matrix metallopeptidase 9 (MMP-9) as potential antigens for developing mRNA vaccines against bladder cancer when analyzing 404 primary bladder cancer tissue samples across four stages I-IV [36]. The discrepancy could be due to the different analysis methods and different patient cohorts in the studies. Nonetheless, these findings warrant further investigation to confirm whether the studied antigens could be TSAs for bladder cancer.

Pancreatic ductal adenocarcinoma (PDAC) exhibits dismal survival rates and demonstrates limited responsiveness to ICI therapy. However, recent studies have suggested that PDAC possesses a greater abundance of neoantigens than previously understood, indicating its potential to activate T cells. Notably, Balachandran et al. observed that tumors characterized by higher counts of neoantigens and increased CD8^+^ T-cell infiltrates were associated with patients experiencing extended survival periods [37]. These neoantigens were notably enriched in the tumor antigen MUC16, also referred to as CA125. Additionally, the researchers designed a model to assess the quality of neoantigens and identified sustained intra-tumoral and circulating T-cell reactivity towards both high-quality neoantigens and MUC16 in long-term survivors of pancreatic cancer. The concept of neoantigen quality holds promise as a potential biomarker for identifying immunogenic tumors, thereby offering insights into guiding the strategic implementation of immunotherapies [37].

In the context of small cell lung cancer (SCLC), Wei et al. conducted a study identifying five potential mRNA vaccine neoantigens, namely, NEK2, NOL4, RALYL, SH3GL2, and ZIC2. Additionally, they delineated two distinct SCLC subtypes categorized as “Immunity High” and “Immunity Low.” Notably, the mRNA vaccine neoantigens displayed increased expression levels in patients characterized by the “Immunity Low” subtype. This observation suggested that the “Immunity Low” subtype might hold greater promise for the development of tumor vaccines while the “Immunity High” subtype might be better suited for treatment utilizing ICI [38]. The identification of suitable neoantigens and the selection of appropriate patients are prerequisites for the mRNA vaccine to work.

## 3. Adjuvant Identification for mRNA Cancer Vaccines

While mRNA vaccines themselves possess the capacity to stimulate the immune system, the inclusion of additional adjuvants has the potential to further amplify vaccine efficacy. Li et al. explored the utilization of synthetic phosphonothioate-modified CPG oligodeoxynucleotides (CPG-ODNs) as an adjuvant in conjunction with an mRNA-based cancer vaccine [39]. Through their investigation, they identified a novel CpG-B class ODN known as CpG2018B, which primarily induced cytokine production via TLR9 pathways. Administering both CpG2018B and an mRNA-encoding neoantigen vaccine intratumorally in melanoma-bearing mice was facilitated by LNPs. Notably, the mRNA vaccine, when using CpG as an adjuvant, exhibited an augmented antitumor effect compared to either CpG or the mRNA vaccine alone. These findings highlighted the possibility of combining CpG with mRNA cancer vaccines, presenting an appealing avenue for the development of immunostimulatory sequence-based therapeutic strategies [39].

Meng and colleagues made a similar noteworthy observation [40]. They engineered a virus-like vaccine particle (VLVP), a composite structure comprising a phospholipid bilayer enveloping a core composed of antigen-encoding mRNA molecules, unmethylated CpG oligonucleotides, and positively charged proteins. Following VLVP treatment, DCs underwent maturation, thereby enhancing antigen presentation [40]. As a result, neoantigen-specific CD8^+^ T cells underwent proliferation in lymphatic organs and effectively infiltrated tumor sites. These findings strongly suggested that CpG oligonucleotides enhanced anti-tumor activity. The inclusion of CpG not only maximized vaccine potency but also exerted a regulatory role by inhibiting PD-1 expression in T cells, thus fulfilling a dual function as a potent adjuvant and a checkpoint blockade agent [40].

In addition to the TLR9 agonist CpG, another promising adjuvant for mRNA cancer vaccines is the TLR4 agonist monophosphoryl lipid A (mPLA). Ma and colleagues engineered an mRNA cancer vaccine which, using mPLA as an adjuvant, demonstrated the ability to induce DC maturation and convert M2 macrophages into M1 macrophages in a mouse model of lung cancer [41]. Consequently, this vaccine effectively suppressed tumor growth and mitigated bone metastasis. The synergy between the mRNA cancer vaccine and the adjuvant mPLA presented compelling insights and potential applications for employing mRNA-based vaccines in the treatment of lung cancer [41].

Apart from TLR agonists, the retinoic acid-inducible gene-I (RIG-I) agonist offers an additional avenue as an adjuvant for mRNA cancer vaccines. Tockary and colleagues innovatively merged antigen-encoding mRNA and an immunostimulatory adjuvant into a singular formulation to heighten the effectiveness of mRNA vaccines [42]. They designed short double-stranded RNA (dsRNA) to target RIG-I, which was then affixed to the mRNA strand via hybridization. Notably, the dsRNA-tethered mRNA, encoding ovalbumin (OVA) and formulated within anionic lipoplexes, elicited substantial therapeutic impacts in a mouse lymphoma (E.G7-OVA) model. Consequently, the integration of an mRNA vaccine and an adjuvant provided a straightforward yet robust platform capable of delivering the desired level of immune stimulation across various mRNA cancer vaccine formulations [42].

## 4. Delivery Materials

An mRNA vaccine should be successfully delivered into target cells in order to perform its function. It first enters the cytoplasm and then is translated into target proteins (Figure 2). For mRNA to gain access to the cytoplasm, it must traverse the negatively charged phospholipid bilayer of the cell membrane. Cells typically permit the entry of only small molecules with a molecular weight of less than 1000 Da via passive diffusion. Therefore, it necessitates a carrier to facilitate mRNA passage into the cell’s cytoplasm. Currently, the predominant delivery method for mRNA vaccines is through the use of LNPs, which require four cumbersome components and suffer from inflammation-related side effects [43]. New delivery methods are warranted and should be extensively investigated.

Fan et al. explored the potential of cationic lipid-assisted nanoparticles (CLANs) for mRNA vaccine delivery [44]. Their findings revealed that the mRNA vaccine administered via CLANs effectively induced DC maturation and triggered the activation of antigen-specific T cells. It is particularly noteworthy that the CLANs encapsulating the mRNA encoding an OVA induced a robust OVA-specific T-cell response, resulting in decelerated tumor growth in a mouse E.G7-OVA lymphoma model. Overall, these results suggested that CLANs hold promise as a robust platform for mRNA vaccine delivery [44].

Lipid/calcium/phosphate nanoparticles (LCP-NPs) were developed by Huang and colleagues [45,46]. The LCP surface was engineered with mannose, a ligand that targets mannose receptors expressed on DCs. This modification facilitated the delivery of exogenous antigens/mRNA into the cytosol of DCs, thus triggering an MHC-I-restricted cytotoxic T lymphocyte response [47]. In a murine 4T1 BC model, the delivery of an mRNA vaccine encoding the tumor antigen MUC1 via LCP-NPs led to the activation and expansion of neoantigen-specific T cells [48]. Remarkably, combination therapy involving mRNA vaccines and anti-CTLA-4 monoclonal antibodies significantly enhanced the anti-tumor immune response when compared to either the vaccine or the monoclonal antibody used alone. These findings highlighted the dual role of LCP-NPs as a carrier for mRNA vaccine delivery and the potential for a combination immunotherapy approach using LCP-NP-based mRNA vaccines and CTLA-4 inhibitors in the context of triple-negative BC [48].

To facilitate antigen introduction into DCs, Tateshita et al. devised a lipoplex-type mRNA carrier utilizing a blend of an SS-cleavable and pH-activated lipid-like material with vitamin E scaffolds (ssPalmE) and the alpha-helical cationic peptide “KALA” (ssPalmE-KALA) [49]. The delivery of mRNA vaccines via the ssPalmE-KALA resulted in markedly heightened protein expression and the secretion of proinflammatory cytokines from murine bone marrow-derived DCs (BMDCs), surpassing the response elicited by a lipoplex crafted with an ssPalm containing fatty acid scaffolds. Notably, immunization with BMDCs previously transfected with an mRNA encoding OVA triggered OVA-specific cytotoxic T-lymphocyte responses in a mouse E.G7-OVA lymphoma model. Collectively, the ssPalmE-KALA approach displayed potent potential as an ex vivo DC-based RNA vaccine platform [49].

The effective delivery of mRNA to specific organs remains a significant challenge in the in vivo application of mRNA technology. Targeted mRNA vaccine delivery to the lymph nodes (LNs) holds the promise of reducing side effects and enhancing the immune response. Chen et al. explored an endogenously LN-targeting LNP named 113-O12B, without modifying any active targeting ligand modifications, for the development of an mRNA-based cancer vaccine [50]. Notably, 113-O12B exhibited heightened and specific expression in the LNs compared to LNPs formulated with ALC-0315, a synthetic lipid used in COVID-19 vaccines. Vaccination with the mRNA-encoding OVA via 113-O12B produced protective and therapeutic effects in a B16F10-OVA melanoma model [50].

Regular B16F10 melanoma displays metastatic behavior and exhibits poor immunogenicity. Tyrosinase-related protein-2 (TRP2), a TAA, is frequently overexpressed in murine and human melanomas, but it possesses poor immunogenicity. As a result, inducing TRP2-specific T-cell immunity has become imperative for generating a robust anticancer immune response when utilizing a TRP2-based cancer vaccine [51]. Remarkably, 113-O12B encapsulating mRNA encoding the TRP-2 peptide exhibited substantial tumor growth inhibition, achieving a complete response rate of 40% in the regular B16F10 melanoma model when combined with anti- PD-1 therapy [50]. These findings emphasized the wide-ranging application of 113-O12B, spanning from protein (OVA) to peptide antigens (TRP2 peptide). Moreover, all the treated mice demonstrated durable immune memories, which prevented the formation of tumor metastatic nodules in subsequent rechallenge experiments involving the lungs. The heightened antitumor efficacy demonstrated by the LN-targeting LNP system underscored its substantial potential as a universal platform for the next generation of mRNA vaccines [50].

LNP systems can provoke substantial inflammation [43]. To address this, Huang and colleagues devised a delivery system that exhibited minimal inflammatory side effects in vivo. In their study, they synthesized a range of alternating copolymers featuring ortho-hydroxy tertiary amine (HTA) repeating units, collectively referred to as PHTA [52]. PHTA serves the triple purpose of condensing mRNA, bolstering polymeric nanoparticle (PNP) stability, and prolonging circulation time. Remarkably, the leading contender among these, PHTA-C18, effectively delivered mRNA cancer vaccines and subsequently triggered a robust CD8^+^ T-cell-mediated antitumor cellular immune response in a mouse B16 melanoma model. This integrated PNP, built upon the foundation of PHTA, represents a promising avenue for developing mRNA cancer vaccines with favorable inflammatory safety profiles [52]. Whether the finding of PHTA can translate into clinical settings warrants further investigation.

## 5. Bringing Together CAR-T Cells and mRNA Cancer Vaccines: A Synergistic Approach

The convergence of CAR-T cells and mRNA cancer vaccines represents a promising development in the field of cancer immunotherapy. CAR-T cell therapies have shown tremendous promise in treating hematological cancers and lymphomas. These therapies involve genetically modifying a patient’s T cells to express CARs that target specific cancer antigens. Despite their success, CAR-T cell therapies face manufacturing challenges. The production processes are often time-consuming, difficult to scale, and expensive. This can limit their widespread availability and affordability. mRNA technology provides a potential solution to overcome these CAR-T cell manufacturing challenges. mRNA encoding CARs is directly delivered into T cells, where it is expressed on the T cell’s surface. This approach allows for the efficient and cost-effective generation of CAR-T cells within a patient’s own body. It eliminates the need for complex ex vivo cell manipulation and expansion, streamlining the therapy’s production. This convergence holds promise for expanding the application of CAR-T cell therapies beyond hematological cancers and lymphomas, potentially opening up new avenues for the treatment of solid tumors.

Billingsley and colleagues developed LNPs designed for delivering mRNA to human T cells ex vivo [53]. In their in vitro study, they identified the most effective LNP formulation, known as C14-4, which was selected for the delivery of CAR mRNA. This platform successfully triggered CAR expression levels comparable to those achieved through electroporation, markedly decreasing cytotoxicity. CAR-T cells engineered using the C14-4 LNP method were subsequently compared to electroporated CAR-T cells, and both methods for CAR-T cell engineering exhibited potent tumor-killing capacities in a coculture experiment with Nalm-6 acute lymphoblastic leukemia cells.

Several studies have emerged demonstrating that an mRNA-based injection can effectively elicit CAR-T cells within the body. In one report, mRNA was utilized to engineer T cells into functional CAR-T cells in vivo [54]. These engineered cells successfully rescued cardiac function in a mouse model of heart failure. In the context of cancer, Parayath and colleagues introduced an injectable nanocarrier designed for delivering mRNA encoding CAR or TCR. This approach enabled the transient reprogramming of circulating T cells to recognize tumor antigens [55]. In mouse models of prostate cancer, hepatitis B-induced hepatocellular carcinoma, and leukemia, repeated infusions of these polymer nanocarriers led to the generation of sufficient host T cells expressing tumor-specific CARs or virus-specific TCRs, resulting in disease regression comparable to infusions of CAR-T cells [55]. Altogether, this convergence of CAR-T cells and mRNA cancer vaccines holds great promise for the immunotherapy of solid tumors.

CRISPR, a cutting-edge gene-editing tool, is revolutionizing cancer research and treatment [56]. One of its remarkable applications involves using CRISPR/Cas-based gene editing to enhance CAR-T cell therapy. In particular, researchers have initiated the first human trial where CRISPR/Cas technology has been employed to delete the PD-1 gene locus in CAR-T cells [57], rendering them resistant to exhaustion. Ren et al. have harnessed CRISPR/Cas to create gene-disrupted allogeneic T cells [58]. These cells are uniquely engineered to lack TCR, HLA class I molecules, and PD-1. This triple-deficient configuration serves to reduce alloreactivity and eliminate the risk of graft-versus-host disease. Moreover, these gene-edited allogeneic T cells offer the potential to serve as universal donor cells [58], bypassing the complexities and high production costs associated with autologous T cells. By introducing mRNA-encoded CARs into these gene-disrupted allogeneic T cells, it is conceivable that we could achieve enhanced efficacy in treating solid tumors. This innovative approach holds significant promise for the future of cancer therapy.

## 6. Clinical Trials of Current mRNA Cancer Vaccines

Multiple clinical trials have indicated that T cells that recognize neoantigens are present in most cancers and these neoantigens provide highly immunogenic and specific targets for individualized vaccination. Currently, numerous ongoing clinical trials exploring mRNA cancer vaccines have gained momentum (Table 1), largely influenced by the success of mRNA-based COVID-19 vaccines. One notable clinical phase I trial (clinicaltrials.gov identifier NCT04161755), reported in the journal *Nature*, unveiled that personalized mRNA vaccines hold the potential to activate neoantigen-specific T cells, thereby delivering clinical benefits for patients with PDAC [59]. The trial intervention encompassed a sequential regimen, involving the administration of an anti-PD-L1 antibody, an individualized mRNA neoantigen vaccine containing up to 20 neoantigens encapsulated in nanoparticles, and a four-drug chemotherapy regimen for individuals with surgically resectable PDAC. Among the recipients of the vaccine, eight out of sixteen patients (50%) generated robust vaccine-specific T-cell responses. Notably, the vaccine-induced T-cell clones in the blood, characterized by a polyfunctional CD8^+^ effector phenotype, displayed no overlap with the anti-PD-L1-induced clones [59,60]. Significantly, the presence of T cells expanded by the vaccine was correlated with a delayed recurrence of PDAC, underscoring the potential of this approach and warranting larger-scale trials for further investigation.

In a clinical phase I trial (NCT03480152), a distinctive approach was adopted where validated neoantigens, predicted neoepitopes, and mutations of driver genes were consolidated into a singular mRNA construct for the vaccination of patients with metastatic gastrointestinal cancer [61]. The trial results emphasized the safety of the mRNA vaccine while also demonstrating its ability to evoke mutation-specific T-cell responses targeted at predicted neoepitopes that were undetectable before vaccination. Furthermore, T-cell receptors designed to target the KRASG12D mutation were detected. However, it is noteworthy that no objective clinical responses were observed among the four patients subjected to this trial [61]. To unlock the full potential of these mRNA vaccines, future avenues involve exploring their combination with ICIs or ACT therapy. These approaches warrant thorough evaluation to determine their potential clinical benefits for patients afflicted with common epithelial cancers.

A multicenter, open-label, two-arm clinical phase I trial (NCT03164772) was conducted to study the safety and efficacy of an mRNA vaccine addition to one or two ICIs in the treatment of non-small cell lung cancer (NSCLC). Arm A was an mRNA vaccine (BI 1361849) plus an anti-PD-L1 antibody and arm B was BI 1361849 plus anti-PD-L1 and anti-CTLA-4 antibodies. The results showed that at the eighth week after treatment, the progression-free survival (PFS) rate was 47.8% and 32.4% in the arm A and arm B cohorts, respectively. At the 24th week after treatment, the PFS rate was 43.5% and 8.8% in arm A and arm B, respectively, indicating that the combination therapy of mRNA vaccines with the anti-PD-L1 antibody displayed better efficacy.

A recent collaboration has emerged between a Claudin18.2-specific CAR-T cell product candidate [62] and Moderna’s investigational Claudin18.2 mRNA cancer vaccine [31]. The CAR-T cell product, known as CT041, was developed by CARsgen, which focuses on autologous CAR-T cell therapies for potential treatment in gastric, pancreatic, and specific digestive system cancers. Claudin18.2 is highly expressed in multiple cancers, making it a potential target for anti-tumor therapy. CT041 is currently undergoing various clinical investigations in both China and North America. Moderna is advancing an investigational off-the-shelf mRNA cancer vaccine designed to encode the Claudin18.2 protein. The trial aims to evaluate the combined potential of CT041 and Moderna’s Claudin18.2 mRNA cancer vaccine, thereby exploring new dimensions for potential treatments.

## 7. Challenges and Future Prospects

During the COVID-19 pandemic, the licensing of two mRNA vaccines played a pivotal role in the immune response to SARS-CoV-2, saving millions of lives. The success of these two vaccines not only effectively addressed the pandemic but also instilled substantial optimism for the future of mRNA cancer vaccines. This achievement has accelerated mRNA vaccine technology and infused vitality into the biopharmaceutical industry.

Despite these successes, there remain areas in mRNA vaccine technology that require refinement. Firstly, a comprehensive exploration of more TAAs or TSAs is essential. Single-neoantigen tumor vaccines may inadvertently permit tumor variants to evade immune responses. To solve this problem, an mRNA vaccine should target multiple TAAs or TSAs simultaneously to enhance therapeutic efficacy. In some clinical trials, personalized vaccines have contained up to 20 neoantigens [59], which might promote vaccine efficacy. It should also be noted that not all cancer patients are suitable for one mRNA vaccine treatment. Therefore, the selection of suitable patients is another factor that influences mRNA vaccine efficacy.

Secondly, in vivo delivery of mRNA vaccines is inefficient. Thus, the development of delivery platforms or materials for mRNA vaccines requires further investigation. While current LNP technology demonstrates effectiveness, the potential for inflammation-related side effects underscores the need for safer and more efficient delivery materials [43]. Ideal materials should precisely target APCs, particularly DCs, triggering robust humoral and cellular immunity. Recent delivery materials targeting specific organs that can reduce side effects require extensive investigation. 

Thirdly, cancer cells suppress the immune system by restricting neoantigen recognition and upregulating the expression of inhibitory B7 family members. To solve this problem, an ideal adjuvant would further enhance the mRNA vaccine-mediated immune response. 

Fourthly, enhancing the stability for formulated mRNA vaccines at ambient temperatures remains a challenge in terms of widespread distribution. Resolving this challenge holds the potential to significantly propel the utility of mRNA vaccines. Concurrently, optimization in the manufacturing processes of mRNA vaccines is necessary.

Fifthly, mRNA vaccines themselves cannot kill larger or advanced tumors. Combination therapy would have big chances to solve this. Numerous ongoing clinical trials are examining the synergy between mRNA vaccines and other treatments such as CAR-T cell or ICI therapies. Currently, the primary resistance mechanisms of anti-PD-1/PD-L1 therapy in solid tumors have been investigated, including insufficient antigen immunogenicity, irreversible T-cell exhaustion, the dysfunction of antigen presentation, resistance to IFN-γ signaling, and suppressive TME [63]. Some oncogenic signaling pathways, such as PTEN loss [64], KRAS mutation [65], EGFR mutations, and ALK rearrangements [66], are also responsible for primary resistance. The combination therapies aim to further augment the efficacies of mRNA cancer vaccines and ICIs. Additionally, convergence of mRNA vaccines and CAR-T cell therapy could enable the in-situ generation of CAR-T cells inside the body. This approach could reduce manufacturing complexity and improve their efficacy against solid tumors. Notably, TME is very suppressive. Whether combination-therapy-induced neoantigen-specific CD8^+^ T cells could efficiently infiltrate into the TME and maintain functional capacity warrants extensive investigation. These results are anticipated with great desire.

Sixthly, people who are older or bearing advanced tumors have weak immune responses due to the disease itself and its treatments, such as chemotherapy or radiation therapy. This can potentially impact their ability to generate a robust immune response to vaccines. However, research has shown that many older cancer patients can still mount a significant immune response to COVID-19 mRNA vaccines, albeit with some variability. Thus, an extensive series of clinical trials is necessary to comprehensively evaluate the immunogenicity and safety aspects of mRNA vaccines in these populations. Such trials are essential to gain a thorough understanding of the potential of mRNA technology. In addition, it becomes necessary to provide more information and vaccination strategies for these populations, enabling the selection of the most suitable biologics based on individual characteristics.

In conclusion, mRNA constitutes a potent and versatile platform for cancer vaccines. An ideal mRNA vaccine formulated in suitable nanoparticles and administered with suitable adjuvants should contain multiple neoantigens that encompass validated neoantigens, predicted neoantigens, and mutations of driver genes. An ideal mRNA vaccine should also target DCs, the strongest antigen-presenting cells. Its successful translation to clinical applications is poised to significantly bolster our capacity to combat cancer. 

## Figures and Tables

**Figure 1 vaccines-11-01599-f001:**
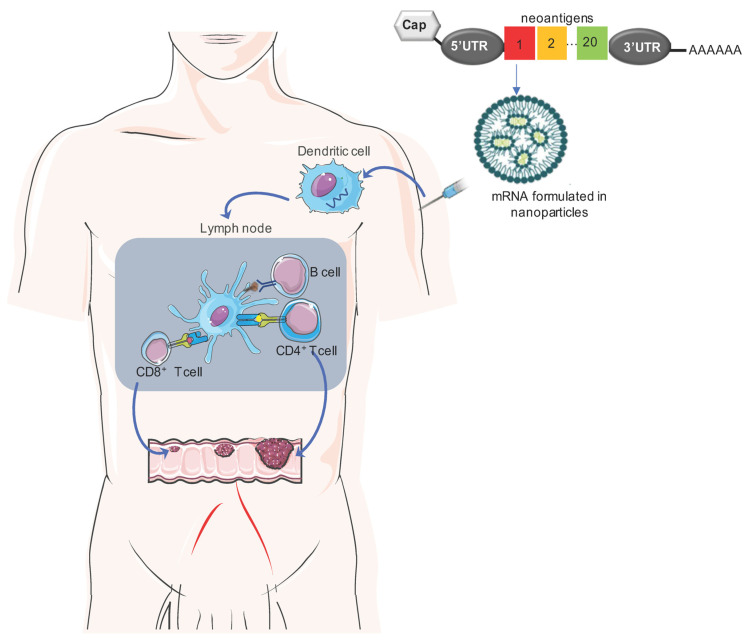
Schematic overview of the mechanism of action of mRNA cancer vaccines: An optimal mRNA vaccine should comprise a variety of neoantigens, including validated neoantigens, predicted neoantigens, and mutations in driver genes, which can collectively reduce the potential for tumor evasion. Upon delivery via nanoparticles, the mRNA vaccine actively targets and activates antigen-presenting cells, such as DCs, through interactions with TLR-7 or TLR-8. Subsequently, mature DCs migrate to lymph nodes, initiating robust B- and T-cell immune responses.

**Figure 2 vaccines-11-01599-f002:**
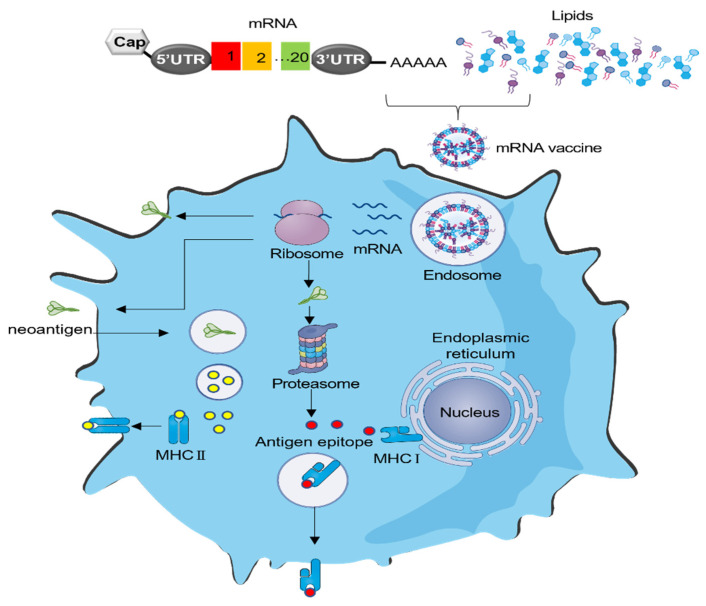
Schematic overview of the delivery and translation mechanism of mRNA cancer vaccines. All mRNA delivery materials incorporate ionizable or cationic molecules, facilitating their passage through the negatively charged phospholipid bilayer of the cell membrane. Upon entry into the cells, the mRNA is translated into the target proteins by the ribosomes. Subsequently, the antigenic proteins can either be localized on the cell’s surface or undergo processing and presentation through MHC–peptide complexes.

**Table 1 vaccines-11-01599-t001:** Current mRNA-based cancer vaccines in clinical trials.

Cancer Type	NCT Number	Study Title	Study Status	Study Results	Interventions	Phases
Solid malignancies	NCT05198752	A study of neoantigen mrna personalised cancer in patients with advanced solid tumors	recruiting	NO	Personalized neoantigen mRNA cancer vaccine	I
	NCT05359354	Safety and efficacy of personalized neoantigen vaccine in advanced solid tumors	recruiting	NO	Personalized neoantigen tumor vaccine	N/A
	NCT05579275	Evaluate the safety and tolerability of jcxh-212 injection in the treatment of advanced malignant solid tumors	recruiting	NO	Neoantigen mRNA cancer vaccine	I
	NCT05714748	Application of mRNA immunotherapy technology in epstein-barr virus-related refractory malignant tumors	recruiting	NO	EBV mRNA vaccine	I
	NCT05916248	Personalized tumor vaccines and pembrolizumab in patients with advanced solid tumors	recruiting	NO	Personalized neoantigen tumor vaccine	I
	NCT05940181	A safety and efficacy study of xh001 combined with sintilimab injection in advanced solid tumors	recruiting	NO	Neoantigen mRNA cancer vaccine with sintilimab	N/A
	NCT05942378	A study of hrxg-k-1939 and adebrelimab in patients with advanced solid tumors	not yet recruiting	NO	Personalized neoantigen tumor vaccine and Adebrelimab	I
	NCT05949775	Clinical study of mRNA vaccine in patients with advanced malignant solid tumors	not yet recruiting	NO	Neoantigen mRNA personalized cancer vaccine	N/A
Gastrointestinal malignancies	NCT03468244	Clinical study of personalized mRNA vaccine encoding neoantigen in patients with advanced digestive system neoplasms	unknown	NO	Personalized mRNA tumor vaccine	N/A
	NCT04161755	Study of personalized tumor vaccines and a pd-l1 blocker in patients with pancreatic cancer that can be treated with surgery	active, not recruiting	YES	Atezolizumab, personalized tumor vaccines, mFOLFIRINOX	I
	NCT05192460	Safety and efficacy of personalized neoantigen vaccine in advanced gastric cancer, esophageal cancer and liver cancer	recruiting	NO	Neoantigen tumor vaccine with or without PD-1/L1	N/A
	NCT05456165	Study of an individualized vaccine targeting neoantigens in combination with immune checkpoint blockade for patients with colon cancer	terminated	NO	Individualized neoantigen tumor vaccine with or without PD-1/L1	II
	NCT05738447	Application of mRNA immunotherapy technology in hepatitis b virus-related refractory hepatocellular carcinoma	recruiting	NO	HBV mRNA vaccine	I
	NCT05761717	Clinical study of mRNA vaccine in patients with liver cancer after operation	not yet recruiting	NO	Neoantigen mRNA personalized cancer vaccine in combination with Stintilimab	N/A
	NCT05916261	Personalized tumor vaccines and pabolizumab in patients with advanced pancreatic cancer	recruiting	NO	Personalized neoantigen tumor vaccine	I
	NCT05981066	A clinical study of mRNA vaccine (abor2014/ipm511) in patients with advanced hepatocellular carcinoma	recruiting	NO	Neoantigen vaccine, I.M injection	N/A
	NCT06019702	Clinical study of personalized mRNA vaccine encoding neoantigen alone in subjects with advanced digestive system neoplasms	recruiting	NO	iNeo-Vac-R01	I
Prostate cancer	NCT02140138	An open label randomised trial of rnactive cancer vaccine in high risk and intermediate risk patients with prostate cancer	terminated	NO	mRNA vaccines (targeting 6 prostate-specific antigens) with needle-free injection device	II
	NCT04382898	Pro-merit (prostate cancer messenger RNA immunotherapy)	recruiting	NO	mRNA vaccines (targeting 5 antigens) with Cemiplimab	I/II
NSCLC	NCT03164772	Phase 1/2 study of combination immunotherapy and messenger ribonucleic acid (mRNA) vaccine in subjects with NSCLC	completed	YES	An mRNA vaccine with or without Durvalumab and Tremelimumab	I/II
Melanoma	NCT05264974	Novel RNA-nanoparticle vaccine for the treatment of early melanoma recurrence following adjuvant anti-pd-1 antibody therapy	not yet recruiting	NO	Autologous tumor mRNA loaded DOTAP liposome vaccine	I
Glioblastoma	NCT05938387	Safety and tolerability of cvgbm in adults with newly diagnosed mgmt-unmethylated glioblastoma or astrocytoma	recruiting	NO	Different doses of an mRNA vaccine	I
Pulmonary Osteosarcoma	NCT05660408	Study of RNA-lipid particle (RNA-LP) vaccines for recurrent pulmonary osteosarcoma (OSA)	not yet recruiting	NO	RNA-LP vaccine	I/II
Melanoma, Gastrointestinal, Genitorinary	NCT03480152	Messenger mRNA)-based, personalized cancer vaccine against neoantigens expressed by the autologous cancer	terminated	YES	National Cancer Institute (NCI)-4650, an mRNA-based, personalized cancer vaccine	I/II
Esophageal cancer and NSCLC	NCT03908671	Clinical study of personalized mRNA vaccine encoding neoantigen in patients with advanced esophageal cancer and non-small cell lung cancer	recruiting	NO	Personalized mRNA tumor vaccine	N/A
NSCLC, pancreatic cancer, and colorectal cancer	NCT03948763	A study of mRNA-5671/v941 as monotherapy and in combination with pembrolizumab (v941-001)	completed	NO	mRNA 5671 with pembrolizumab	I

Note: N/A, Not Applicable.

## Data Availability

Not applicable.

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
