# Peer review of "Advances in mRNA-Based Cancer Vaccines"

_vaccines, 2023, doi:10.3390/vaccines11101599_

Round 1

Reviewer 1 Report

The review article entitled ‘Advances in mRNA-based cancer vaccines’ is very intriguing, well written, and embodies latest advancements in the field. The author could improve the article.

1.     The introduction about neoantigen therapy is too short, they can add some more details

2.     An overall summary picture of the mechanism of targeting of neoantigens would be a great addition to the review.

3.     The authors need to describe about mRNA vaccines that were FDA approved,

4.     The description of delivery mechanism in more detail, and an illustrative picture is required.

5.     The description of different types of breast cancer is required, the proposed antigens are to target all different type of BC or only a specific type. Some more information need to be provided.

6.     Line 151, the author needs to quote a reference to the statement.

7.     The authors can discuss about the resistance mechanisms of PD-1 and some of the pathways related to it.

8.     The authors can include a separate section of the emerging targets that could be potentially explored in the field such as mRNA vaccines.

9.     The author can describe about the potential use of CRISPR-CAS9 for cancer vaccines.

10.  The author need to include discussion and discuss more broadly.

Author Response

Dear Reviewer,

We are grateful to the reviewer for the constructive feedback and valuable suggestions. We have carefully considered each point raised and have made the necessary revisions to enhance the quality of our review article. Below, we provide a detailed response to each comment.

Comment #1:  The introduction about neoantigen therapy is too short, they can add some more details.

Response: We appreciate this suggestion. We have added more descriptions of neoantigen therapy in the revised text (lines 55-59).

Comment #2: An overall summary picture of the mechanism of targeting of neoantigens would be a great addition to the review.

Response: Thanks for your suggestion. We have revised Figure 1 and put it in the section of “1. mRNA cancer vaccines and their mechanism of action” (page 4).

Comment #3: The authors need to describe about mRNA vaccines that were FDA approved.

Response: Thanks for your suggestion. We have added the description of the FDA-approved mRNA vaccines (mRNA-1273 and BNT162b2) in the revised version, as suggested by the reviewer (lines 125-128).

Comment #4: The description of delivery mechanism in more detail, and an illustrative picture is required.

Response: We highly appreciate this suggestion. We have added the descriptions of delivery mechanism in the revised text (lines 288-293) and a new Figure 2 that illustrates delivery mechanism (page 7).

Comment #5: The description of different types of breast cancer is required, the proposed antigens are to target all different type of BC or only a specific type. Some more information need to be provided. 

Response: Thank you for pointing it out. We have added the description of different subtypes of breast cancer (lines 186-189), as suggested by the reviewer. In addition, we also added more detailed about the BC samples (lines 191-192).

Comment #6: Line 151, the author needs to quote a reference to the statement.

Response: We apologize for the oversight. We have added the reference as suggested by the reviewer (line 202).

Comment #7: The authors can discuss about the resistance mechanisms of PD-1 and some of the pathways related to it. 

Response: We highly appreciate this suggestion. In the section of “7. Challenges and Future Prospects”, we have discussed the resistance mechanisms of anti-PD-1/PD-L1 therapies (lines 511-517).

Comment #8: The authors can include a separate section of the emerging targets that could be potentially explored in the field such as mRNA vaccines. 

Response: Thanks for your suggestion. The new section of “5. Bringing Together CAR-T Cells and mRNA Cancer Vaccines: A Synergistic Approach” has been added in the revised text (lines 372-422), as suggested by the reviewer.

Comment #9: The author can describe about the potential use of CRISPR-CAS9 for cancer vaccines. 

Response: Thanks for your suggestion. We have added a description of CRISPR-CAS9 and its application in the revised text (lines 410-422).

Comment #10: The author need to include discussion and discuss more broadly.

Response: Thanks for your suggestion. In the section of “7. Challenges and Future Prospects” of this review article, we have discussed more broadly (lines 508-534).

We value the reviewer’s feedback, which has guided us in refining our manuscript. We believe that the revisions made have addressed the concerns raised and have further enhanced the quality of our work. We look forward to any additional feedback and hope for the favorable consideration of our manuscript for publication.

Sincerely,

Reviewer 2 Report

This is a very well written review and I have no suggested correction or concern with respect to content of this manuscript, my only suggestions are:

1. Provide a detailed schematic representation to demonstrate the mode of action for mRNA vaccine as described in Section 1.2. Figure 1 could be used as a graphical abstract or in the introduction.

2. Incorporation of some figures related to efficacy of mRNA vaccines would increase the value of this paper. 

3. This is a very niche field, I think this paper encompassed the recent advancements pretty well. 

Author Response

Dear Reviewer,

We are grateful to the reviewer for the constructive feedback and valuable suggestions. We have carefully considered each point raised and have made the necessary revisions to enhance the quality of our review article. Below, we provide a detailed response to each comment.

Comment #1: Provide a detailed schematic representation to demonstrate the mode of action for mRNA vaccine as described in Section 1.2. Figure 1 could be used as a graphical abstract or in the introduction.

Response: We highly appreciate this suggestion. We have revised Figure 1 and put it in the section of “1. mRNA cancer vaccines and their mechanism of action” (page 4). In addition, we also added new Figure 2 to show the delivery and translation process of mRNA vaccines in the section of “4. Delivery materials” (page 7).

Comment #2: Incorporation of some figures related to efficacy of mRNA vaccines would increase the value of this paper. 

Response: Great thanks! However, to incorporate the figures related to efficacy of mRNA vaccines, we must obtain permission from the owner of the copyright of the original figure. It is time-consuming. Thus, we did not add those figures.

Comment #3: This is a very niche field, I think this paper encompassed the recent advancements pretty well. 

Response: Thanks for your suggestion. After addressing all your comments, we think the quality of this review article has been significantly improved.

We value the reviewer’s feedback, which has guided us in refining our manuscript. We believe that the revisions made have addressed the concerns raised and have further enhanced the quality of our work. We look forward to any additional feedback and hope for the favorable consideration of our manuscript for publication.

Sincerely,

Reviewer 3 Report

Although the review is pretty useful, there are some points to correct. The major point is section  1.5 regarding current clinical trials. There are no important info about antigens, doses,  regimens, forms/adjuvant, patient selection, troal phase etc.  For instance  column "conditions" just repeat study title. It looks loke for the authors clinical trials do not play much role, at least in his review.

Minor points:

L78-79 - should be low "efficacy"

L140 - Breast cancers are not of poor prognosis, only the triple-negative BC

Good

Author Response

Dear Reviewer,

We are grateful to the reviewer for the constructive feedback and valuable suggestions. We have carefully considered each point raised and have made the necessary revisions to enhance the quality of our review article. Below, we provide a detailed response to each comment.

Comment #1: Although the review is pretty useful, there are some points to correct. The major point is section  1.5 regarding current clinical trials. There are no important info about antigens, doses,  regimens, forms/adjuvant, patient selection, troal phase etc.  For instance  column "conditions" just repeat study title. It looks like for the authors clinical trials do not play much role, at least in his review.

Response: We highly appreciate this suggestion. In the revised version, we deleted “conditions”, and added “study results, trial phase, and intervention” to provide more details for the clinical trials in Table 1.

Comment #2: L78-79 - should be low "efficacy"

Response: We apologize for the oversight. We have made changes, as suggested by the reviewer (line 90).

Comment #3: L140 - Breast cancers are not of poor prognosis, only the triple-negative BC.

Response: Thank you for pointing it out. The sentence was rewritten to “Triple-negative BC patients received poor prognoses and it is resistant to chemotherapy” (line 189).

We value the reviewer’s feedback, which has guided us in refining our manuscript. We believe that the revisions made have addressed the concerns raised and have further enhanced the quality of our work. We look forward to any additional feedback and hope for the favorable consideration of our manuscript for publication.

Sincerely,

Reviewer 4 Report

Comments on advances in mRNA cancer vaccines:
It is a very interesting topic. The manuscript is potentially approvable provided the following comments are addressed:

1. Line 44 remove indeed-cancer indeed is a serious and complex disease.
2. Lines 87/88 Provenge faded out of the market due to its efficacy and high price. Do you mean low efficacy? If so, add clarification word about what is meant by efficacy. Also it would be nice if you are able to validate this by stating the OS/PFS/RR achieved by provenge e.g. with OS of…X number of months, this represents an absolute increase of Z months …..from other available therapies
3. Line 98 unstable not instable.
4. Line 98 it is not clear what is meant by unstable and inefficient in vivo delivery, delivery of what? or delivery of the vaccine to target cells?
5. Line 140: BC are of poor prognosis, resistance to chemotherapy… this is not true as not all BC types are of poor prognosis resistance …. Some have good prognosis and respond to chemotherapy, hormonal therapy, and radiotherapy, etc…Please be more specific.
6. Line 151: confirmed in a large cohort or large study. Also, which type of BC we are talking about here?
7. 162-167 what type of bladder cancer? Are we talking about stage 0a, 0is, or more advanced stages.
8. Line 220-when adjuvanted-strange way to describe adjuvant therapy? Change to hen given as adjuvant therapy.
9. Line 211: oligonucleotides are indispensable-that means cannot function with it. Is this the case or it is recommended as it enhances anti-tumor activity.
10. Line 217 avoid adjuvanted.
11. Line 230/231: this system introduces a straightforward yet robust…What is the meaning of this?
12. Line 268: delivery of mRNA to specific organs remains a challenge. It is nice that this is mentioned here, however before discussing all this (may be in the paragraph before 1.1) you need to mention the general challenges for the treatment with cancer vaccines e.g., the suppression of cancer cells to immune system. Cancer cells may not be perceived as harmful cells, etc. The challenges need to be presented in more details as they represent a major roadblock for the development and treatment with cancer vaccines.
13. Recommend checking about this on Cancer.Net website:
https://www.cancer.net/navigating-cancer-care/how-cancer-treated/immunotherapy-and-vaccines/what-are-cancer-vaccines
Or you can refer to some publications e.g.,
Potentialities and challenges of mRNAin cancer immunotherapy-Duan et al, 2022
https://www.cancer.net/navigating-cancer-care/how-cancer-treated/immunotherapy-and-vaccines/what-are-cancer-vaccines
14. Table 1 needs to be better organized with each cell and column wording is clearly separated from the others. Suggest separating columns and rows by lines. It needs better organization with clinical trials in each disease combined e.g., Melanoma studies, GI studies. Likewise, it is not sufficient to present all these studies without any information e.g., for the ones that have finished, any results presented for them? When to expect some data from the ongoing ones.
15. Line 355 personized? Mean personalized?

16. Challenges and future page 11: this covers some challenges, however, not all of them. A key challenge is the treatment of large or advanced tumors with cancer vaccines. The issue of stimulating the immune system in the elderly, etc.-refer to point 13 above.
17. From what is written, it is not clear what are the prospects? What are the plans to overcome challenges? What are the ongoing activities to address these challenges? Please add some details.
18. General comment, please avoid repeating the word underscores., you can use other words to state the importance of the results of a research activity.
19. Figure 1: this is misplaced. It is better to be moved before the text 1.1. or 1.2
20. General comment, it might be better if the author applies for professional English editing (recommended not mandated).

Fair, recommend to have professional editing (although not mandated).

Author Response

Dear Reviewer,

We are grateful to the reviewer for the constructive feedback and valuable suggestions. We have carefully considered each point raised and have made the necessary revisions to enhance the quality of our review article. Below, we provide a detailed response to each comment.

It is a very interesting topic. The manuscript is potentially approvable provided the following comments are addressed:
Comment #1: Line 44 remove indeed-cancer indeed is a serious and complex disease.

Response: Thank you for pointing it out. We have made changes as suggested by the reviewer (line 48).

Comment #2: Lines 87/88 Provenge faded out of the market due to its efficacy and high price. Do you mean low efficacy? If so, add clarification word about what is meant by efficacy. Also it would be nice if you are able to validate this by stating the OS/PFS/RR achieved by provenge e.g.

Response: We highly appreciate this suggestion. In the revised version, we had added “low efficacy with median overall survival of 25.9 and 21.4 months in the Provenge and Placebo groups, respectively” to make it clear (lines 90-92).

Comment #3: Line 98 unstable not instable.

Response: We apologize for the typo. We have corrected it (line 134).

Comment #4: Line 98 it is not clear what is meant by unstable and inefficient in vivo delivery, delivery of what? or delivery of the vaccine to target cells?

Response: We highly appreciate this suggestion. We have rewritten that sentence to make it clear (line 134).

Comment #5: Line 140: BC are of poor prognosis, resistance to chemotherapy… this is not true as not all BC types are of poor prognosis resistance …. Some have good prognosis and respond to chemotherapy, hormonal therapy, and radiotherapy, etc…Please be more specific.

Response: Thank you for pointing it out. We have added a description of types of breast cancer and revised the text (lines 186-189).

Comment #6: Line 151: confirmed in a large cohort or large study. Also, which type of BC we are talking about here?

Response: We highly appreciate this suggestion. In the revised version, we had added “triple-negative” (line 202).

Comment #7: 162-167 what type of bladder cancer? Are we talking about stage 0a, 0is, or more advanced stages.

Response: Thank you for pointing it out. We have added more details for bladder cancer samples as shown in lines 206-207 and lines 218-219.

Comment #8: Line 220-when adjuvanted-strange way to describe adjuvant therapy? Change to hen given as adjuvant therapy.

Response: Great thanks! We have made changes, as suggested by the reviewer (line 270).

Comment #9: Line 211: oligonucleotides are indispensable-that means cannot function with it. Is this the case or it is recommended as it enhances anti-tumor activity.

Response: Thank you for pointing it out. We have rewritten the sentence to “CpG oligonucleotides enhanced anti-tumor activity” (lines 264-265).

Comment #10: Line 217 avoid adjuvanted.

Response: Great thanks! We have made changes to “when using CpG an adjuvant” (line 254).

Comment #11: Line 230/231: this system introduces a straightforward yet robust…What is the meaning of this?

Response: We apologize for the confusion. We have replaced “this system introduces” with “the integration of a mRNA vaccine and an adjuvant provided” to make it easier to understand (lines 283-284).

Comment #12: Line 268: delivery of mRNA to specific organs remains a challenge. It is nice that this is mentioned here, however before discussing all this (may be in the paragraph before 1.1) you need to mention the general challenges for the treatment with cancer vaccines e.g., the suppression of cancer cells to immune system. Cancer cells may not be perceived as harmful cells, etc. The challenges need to be presented in more details as they represent a major roadblock for the development and treatment with cancer vaccines.

Response: We highly appreciate this suggestion. Cancer cells evade immune surveillance through multiple mechanisms, including restricting neoantigen recognition, suppressing the immune system, and inducing T cell exhaustion, which represent a major roadblock for the development and treatment with cancer vaccines. We have added these descriptions (lines 84-87).

Comment #13: Recommend checking about this on Cancer.Net website:
https://www.cancer.net/navigating-cancer-care/how-cancer-treated/immunotherapy-and-vaccines/what-are-cancer-vaccines
Or you can refer to some publications e.g.,
Potentialities and challenges of mRNAin cancer immunotherapy-Duan et al, 2022
https://www.cancer.net/navigating-cancer-care/how-cancer-treated/immunotherapy-and-vaccines/what-are-cancer-vaccines

Response: We highly appreciate this suggestion. We have discussed more broadly in the section of “7. Challenges and future prospects” (lines 488-534).

Comment #14: Table 1 needs to be better organized with each cell and column wording is clearly separated from the others. Suggest separating columns and rows by lines. It needs better organization with clinical trials in each disease combined e.g., Melanoma studies, GI studies. Likewise, it is not sufficient to present all these studies without any information e.g., for the ones that have finished, any results presented for them? When to expect some data from the ongoing ones.

Response: We highly appreciate this suggestion. We made changes as suggested by the reviewer (pages 10-15). For the clinical trials with study results (NCT04161755, NCT03480152 and NCT03164772), we also described them separately in the section of “6. Clinical trials of current mRNA cancer vaccines” (pages 10 and 15). As to the question “when to expect some data from the ongoing ones”, we have double-checked the website-clinicaltrials.gov, but we could not get those information.

Comment #15: Line 355 personized? Mean personalized?

Response: We apologize for the typo. We have corrected it (line 489).

Comment #16: Challenges and future page 11: this covers some challenges, however, not all of them. A key challenge is the treatment of large or advanced tumors with cancer vaccines. The issue of stimulating the immune system in the elderly, etc.-refer to point 13 above.

Response: We highly appreciate this suggestion. We have made changes in the section of “7. Challenges and future prospects” (lines 524-534). The following paragraph was added.

Sixthly, people who are older or bearing advanced tumors have weak immune responses due to the disease itself and its treatments, such as chemotherapy or radiation therapy. This can potentially impact their ability to generate a robust immune response to vaccines. However, research has shown that many older cancer patients can still mount a significant immune response to COVID-19 mRNA vaccines, albeit with some variability. Thus, an extensive series of clinical trials is necessary to comprehensively evaluate the immunogenicity and safety aspects of mRNA vaccines in these populations. Such trials are essential to gain a thorough understanding of the potential of mRNA technology. In addition, it becomes necessary to provide more information and vaccination strategies for these populations, enabling the selection of the most suitable biologics based on individual characteristics.

Comment #17: From what is written, it is not clear what are the prospects? What are the plans to overcome challenges? What are the ongoing activities to address these challenges? Please add some details.

Response: We highly appreciate this suggestion. We have added more details of future prospects in the section of “7. Challenges and future prospects” (lines 488-534).

Comment #18: General comment, please avoid repeating the word underscores., you can use other words to state the importance of the results of a research activity.

Response: Thanks for your suggestion. We have made changes, as suggested by the reviewer (lines 166, 255, 321, 352 and 451).

Comment #19: Figure 1: this is misplaced. It is better to be moved before the text 1.1. or 1.2

Response: We highly appreciate this suggestion. We have moved it to the section of “1. mRNA cancer vaccines and their mechanism of action” (page 4).

Comment #20: General comment, it might be better if the author applies for professional English editing (recommended not mandated).

Response: Thanks for your suggestion. We had this article checked by a professional editing service. By addressing these issues, we believe the quality of the article has been significantly improved. 

We value the reviewer’s feedback, which has guided us in refining our manuscript. We believe that the revisions made have addressed the concerns raised and have further enhanced the quality of our work. We look forward to any additional feedback and hope for the favorable consideration of our manuscript for publication.

Sincerely,

Reviewer 5 Report

This paper is a summary of a very significant topic; as of now, over 100 mRNA cancer vaccine interventional trials are undergoing.[i] It would be necessary to present an analysis of the trend to conclude the direction. Over 2,600 articles on the topic reported in PubMed[ii] , such as the one below, are not cited in the article.

Wang B, Pei J, Xu S, Liu J, Yu J. Recent advances in mRNA cancer vaccines: meeting challenges and embracing opportunities. Front Immunol. 2023 Sep 6;14:1246682. doi: 10.3389/fimmu.2023.1246682. PMID: 37744371; PMCID: PMC10511650.

There are over 600 systematic reviews, of which 110 were published within the past 12 months. This paper does not bring any new perspective and fails to cite the most important published work.

Zhao J, Liao B, Gong L, Yang H, Li S, Li Y. Knowledge mapping of therapeutic cancer vaccine from 2013 to 2022: A bibliometric and visual analysis. Hum Vaccin Immunother. 2023 Aug;19(2):2254262. doi: 10.1080/21645515.2023.2254262. Epub 2023 Sep 20. PMID: 37728107; PMCID: PMC10512878.

Zhang A, Ji Q, Sheng X, Wu H. mRNA vaccine in gastrointestinal tumors: Immunomodulatory effects and immunotherapy. Biomed Pharmacother. 2023 Oct;166:115361. doi: 10.1016/j.biopha.2023.115361. Epub 2023 Sep 4. PMID: 37660645.

Tojjari A, Saeed A, Singh M, Cavalcante L, Sahin IH, Saeed A. A Comprehensive Review on Cancer Vaccines and Vaccine Strategies in Hepatocellular Carcinoma. Vaccines (Basel). 2023 Aug 12;11(8):1357. doi: 10.3390/vaccines11081357. PMID: 37631925; PMCID: PMC10459477.

Al Fayez N, Nassar MS, Alshehri AA, Alnefaie MK, Almughem FA, Alshehri BY, Alawad AO, Tawfik EA. Recent Advancement in mRNA Vaccine Development and Applications. Pharmaceutics. 2023 Jul 18;15(7):1972. doi: 10.3390/pharmaceutics15071972. PMID: 37514158; PMCID: PMC10384963.

Liu C, Shi Q, Huang X, Koo S, Kong N, Tao W. mRNA-based cancer therapeutics. Nat Rev Cancer. 2023 Aug;23(8):526-543. doi: 10.1038/s41568-023-00586-2. Epub 2023 Jun 13. PMID: 37311817.

Han Y, Shin SH, Lim CG, Heo YH, Choi IY, Kim HH. Synthetic RNA Therapeutics in Cancer. J Pharmacol Exp Ther. 2023 Aug;386(2):212-223. doi: 10.1124/jpet.123.001587. Epub 2023 May 15. PMID: 37188531.

This paper is a superficial incomplete review of a fast maturing technology that is the subject of extensive publication; a new paper must add something useful, such as an analysis of the technology trend with proper citations.

[i] https://clinicaltrials.gov/search?cond=cancer&term=mrna%20%20cancer%20vaccine

[ii] https://pubmed.ncbi.nlm.nih.gov/?term=mrna+cancer+vaccine&sort=date&size=200

Editing

Author Response

Dear Reviewer,

We are grateful to the reviewer for the constructive feedback and valuable suggestions. We have carefully considered each point raised and have made the necessary revisions to enhance the quality of our review article. Below, we provide a detailed response to each comment.

Comment #1: This paper is a summary of a very significant topic; as of now, over 100 mRNA cancer vaccine interventional trials are undergoing.[i] It would be necessary to present an analysis of the trend to conclude the direction.

[I]https://clinicaltrials.gov/search?cond=cancer&term=mrna%20%20cancer%20vaccine

Response: We apologize for the oversight. For the 113 clinical trials mentioned by the reviewer, we have carefully examined them. Among them, 25 were trials on COVID-1 mRNA vaccines in cancer patients (NCT04872738, NCT05270967, NCT04792567, NCT05831787, NCT05025514, NCT05556720, NCT05119855, NCT05672355, NCT05075538, NCT05119738, NCT04847050, NCT04862806, NCT04775056, NCT04932863, NCT05016622, NCT04878796, NCT05028374, NCT04852796, NCT05050461, NCT05597761, NCT04969601, NCT05007860, NCT04918940, NCT04951323, NCT04935528). The majority of the remaining were dendritic cell (DC) vaccines, either loaded with peptides or transfected with mRNA, such as NCT01530698, NCT02465268, NCT01456104 and etc. However, there were still 9 trials on mRNA-based cancer vaccines that we have not previously found. In the revised version, we added those 9 trials in Table 1.

Comment #2: Over 2,600 articles on the topic reported in PubMed[ii] , such as the one below, are not cited in the article.

Wang B, Pei J, Xu S, Liu J, Yu J. Recent advances in mRNA cancer vaccines: meeting challenges and embracing opportunities. Front Immunol. 2023 Sep 6;14:1246682. doi: 10.3389/fimmu.2023.1246682. PMID: 37744371; PMCID: PMC10511650.

There are over 600 systematic reviews, of which 110 were published within the past 12 months. This paper does not bring any new perspective and fails to cite the most important published work.

Zhao J, Liao B, Gong L, Yang H, Li S, Li Y. Knowledge mapping of therapeutic cancer vaccine from 2013 to 2022: A bibliometric and visual analysis. Hum Vaccin Immunother. 2023 Aug;19(2):2254262. doi: 10.1080/21645515.2023.2254262. Epub 2023 Sep 20. PMID: 37728107; PMCID: PMC10512878.

Zhang A, Ji Q, Sheng X, Wu H. mRNA vaccine in gastrointestinal tumors: Immunomodulatory effects and immunotherapy. Biomed Pharmacother. 2023 Oct;166:115361. doi: 10.1016/j.biopha.2023.115361. Epub 2023 Sep 4. PMID: 37660645.

Tojjari A, Saeed A, Singh M, Cavalcante L, Sahin IH, Saeed A. A Comprehensive Review on Cancer Vaccines and Vaccine Strategies in Hepatocellular Carcinoma. Vaccines (Basel). 2023 Aug 12;11(8):1357. doi: 10.3390/vaccines11081357. PMID: 37631925; PMCID: PMC10459477.

Al Fayez N, Nassar MS, Alshehri AA, Alnefaie MK, Almughem FA, Alshehri BY, Alawad AO, Tawfik EA. Recent Advancement in mRNA Vaccine Development and Applications. Pharmaceutics. 2023 Jul 18;15(7):1972. doi: 10.3390/pharmaceutics15071972. PMID: 37514158; PMCID: PMC10384963.

Liu C, Shi Q, Huang X, Koo S, Kong N, Tao W. mRNA-based cancer therapeutics. Nat Rev Cancer. 2023 Aug;23(8):526-543. doi: 10.1038/s41568-023-00586-2. Epub 2023 Jun 13. PMID: 37311817.

Han Y, Shin SH, Lim CG, Heo YH, Choi IY, Kim HH. Synthetic RNA Therapeutics in Cancer. J Pharmacol Exp Ther. 2023 Aug;386(2):212-223. doi: 10.1124/jpet.123.001587. Epub 2023 May 15. PMID: 37188531.

Response: Thank you for bringing these recent publications to our attention. We have now cited the papers in the relevant sections of our manuscript to provide a comprehensive perspective on the topic. The references can be found on page 2 lines 95-99 and page 3 lines 100-115.

Comment #3: This paper is a superficial incomplete review of a fast maturing technology that is the subject of extensive publication; a new paper must add something useful, such as an analysis of the technology trend with proper citations.

[ii] https://pubmed.ncbi.nlm.nih.gov/?term=mrna+cancer+vaccine&sort=date&size=200

Response: We apologize for the oversight. The new section of “5. Bringing Together CAR T Cells and mRNA Cancer Vaccines: A Synergistic Approach” has been added in the revised text (lines 372-422). In addition, we also had this article checked by a professional editing service. By addressing these issues, we believe the quality of the article has been significantly improved.  

We value the reviewer’s feedback, which has guided us in refining our manuscript. We believe that the revisions made have addressed the concerns raised and have further enhanced the quality of our work. We look forward to any additional feedback and hope for the favorable consideration of our manuscript for publication.

Sincerely,

Round 2

Reviewer 3 Report

No comments

No comments

Reviewer 4 Report

I would like to commend and acknowledge the author’s effort to address the comments so diligently. It is now a very good robust, and clear manuscript that provides very useful information. It serves as a reference for future manuscripts addressing the same topic.

Reviewer 5 Report

Sufficient changes have been made as advised.

I see minor grammar errors perhaps they can put it through the MDPI editing